# Analysis of Crack-Characteristic Stress and Energy Characteristics of Sandstone under Triaxial Unloading Confining Pressure

**Yanwei Duan** , **Guohua Zhang * and Tao Qin**

Key Laboratory of Mining Engineering of Heilongjiang Province College, Heilongjiang University of Science and Technology, Harbin 150022, China
* Correspondence: 1994800303@usth.edu.cn

**Abstract:** The deformation and failure of underground engineering are usually caused by unloading. In this work, triaxial unloading confining pressure tests are carried out to simulate the failure process of rock mass caused by unloading, analyze the crack-characteristic stress, and study the energy evolution of rock under unloading and the pre-peak and post-peak energy characteristics combined with the energy theory. The results show that, when the confining pressure increases from 5 MPa to 20 MPa, crack closure stress $\sigma_{cc}$, crack initiation stress $\sigma_{ci}$, dilatancy stress $\sigma_{cd}$, and peak stress $\sigma_p$ are 6.34 times, 2.75 times, 1.93 times, and 1.66 times higher than the original, respectively. By comparing the increase in crack-characteristic stress, it can be found that the confining pressure has a large effect on the crack closure stress and crack initiation stress, while the dilatation stress and peak stress have relatively little influence. From the perspective of energy evolution, the pre-peak axial absorption energy $U_1$ increases exponentially, the elastic energy $U^e$ is similar to $U_1$, and the circumferential consumption energy $U_3$ and dissipation energy $U^d$ are small. After reaching the peak stress, the growth rate of $U_1$ decreases slightly, $U^e$ decreases rapidly, and $U_3$ increases rapidly but only as a small fraction of the total energy, while $U^d$ grows almost exponentially and rapidly becomes the main part of the energy. Under each crack-characteristic stress state, the energy characteristic parameters gradually increase with the increase in confining pressure, which is manifested by the increase in slope in the linear fitting formula of energy characteristic parameters. The release process of the releasable elastic energy after the peak stress can be divided into three stages of "slow–fast–slow", and the energy release process shows an obvious confining pressure effect.

**Keywords:** triaxial unloading confining pressure test; mechanical properties; crack-characteristic stress; energy evolution; energy release

## 1. Introduction

In underground engineering, the stress state of rock mass is usually very complicated under the joint action of geological conditions and mining. To facilitate the analysis and research, the stress state of rock mass is usually simplified and can be understood as a three-dimensional compression state. Due to roadway excavation, the load of surrounding rock is relieved in a certain direction, which makes the original equilibrium state suddenly unstable. To rebalance the stress, the stress will transfer to the depth of the surrounding rock. This transfer process is inevitably accompanied by energy transformation, which leads to the deformation and failure of the surrounding rock. In essence, the deformation and failure of surrounding rock result in the loading and unloading of rock mass caused by unloading. After unloading, the stress in a certain direction of the surrounding rock increases, forming the loading condition, or decreases, forming the unloading condition. Thus, the ultimate bearing capacity of the surrounding rock is limited. Loading will lead to stress exceeding the ultimate bearing capacity, while unloading will reduce the capacity. Under the action of these two aspects, rock mass deformation and failure will occur. Stress

redistribution and strain energy accumulation make the gathered energy of surrounding rock easily released along the working face, resulting in surrounding rock damage, cracking, and even sudden instability of surrounding rock [1–3]. The surrounding rock mass always exchanges energy with the outside world during deformation, damage, and failure [4]. Therefore, how to consider the mechanical properties and energy evolution characteristics of unloading rock mass is the focus of rock deformation and failure mechanism research.

For rock deformation analysis and energy evolution, scholars have carried out rock loading test studies under different test conditions and obtained fruitful results. Li et al. [5] analyzed the influence of cavities and cross-joints on the energy characteristics of rock samples and the energy evolution law by carrying out uniaxial loading tests and combining them with the rock energy theory. Zhao et al. [6] discussed the size effects of energy accumulation and dissipation in sandstone through a uniaxial compression test, and revealed that the elastic energy evolution of sandstone with different aspect ratios follows the law of linear energy storage. Hou et al. [7] carried out uniaxial tests on sandstone with different water content and analyzed the relationship between energy dissipation and the number of cracks in the loading process. Luo et al. [8] analyzed the law of characteristic energy in the loading process by carrying out a uniaxial cyclic loading test of sandstone. Zhang et al. [9] carried out triaxial tests on three kinds of rocks to study energy characteristics such as energy dissipation and energy storage limit during rock deformation. Zhang et al. [10] analyzed the relationship between wave velocity and energy dissipation in the red sandstone deformation and failure process based on the full stress–strain curve by carrying out triaxial loading tests. Dai et al. [11] obtained the transformation law of axial elastic energy and dissipated energy by conducting rock mechanics tests under different stress paths. Yang et al. [12] carried out rock mechanics tests with different loading modes and analyzed the rock deformation and failure mechanism and post-peak energy evolution characteristics. Meng et al. [13] revealed the relationship between energy rebound density and stress by carrying out rock mechanics tests at different loading rates, and then analyzed the microscopic mechanism of energy characteristics. Liu et al. [14] combined with digital image correlation technology and acoustic emission technology, and then conducted direct shear tests on different rough joints to analyze the deformation energy evolution law. Liu et al. [15] obtained the linear energy storage law and creep energy characteristics of red sandstone by conducting creep experiments.

In summary, through rock mechanics tests with different loading methods, scholars have discussed the influencing factors of rock energy and the law of energy evolution. However, there are few studies on the crack-characteristic stress and the corresponding energy characteristics of rock under unloading. In this work, the triaxial unloading confining pressure test is carried out to simulate the failure process of rock mass caused by unloading, analyze the crack-characteristic stress in the deformation and failure process, and analyze the energy evolution of rock under unloading and the energy characteristics combined with the energy theory. The results enrich the research on rock deformation and failure under unloading, have important significance for the study of rock failure mechanism, and can provide guidance for rock excavation and unloading surrounding rock support.

## 2. Test Conditions and Scheme

### 2.1. Test Conditions

The rock samples used in this test were taken from Xinjian Coal Mine in Qitaihe, China. According to the site construction situation, the left fourth working face of 93# Coal was selected to collect the roof rock samples, and the lithology was gray or light-gray sandstone, containing carbonized plant debris. According to the ISRM standard, the rock was cored, cut, and processed into a $\Phi$50 mm $\times$ 100 mm standard specimen. To ensure the homogenization of the samples, the Sonic Viewer-SX ultrasonic wave velocity test system was used to select samples with a wave velocity of about 2100 m/s. This batch of rock samples was used for the subsequent tests.

The test equipment was a TOP INDUSTRIE Rock 600-50 Rock automatic servo rheometer (see Figure 1), which consists of axial, confining, and seepage pressure servo devices to obtain both axial and circumferential strains.

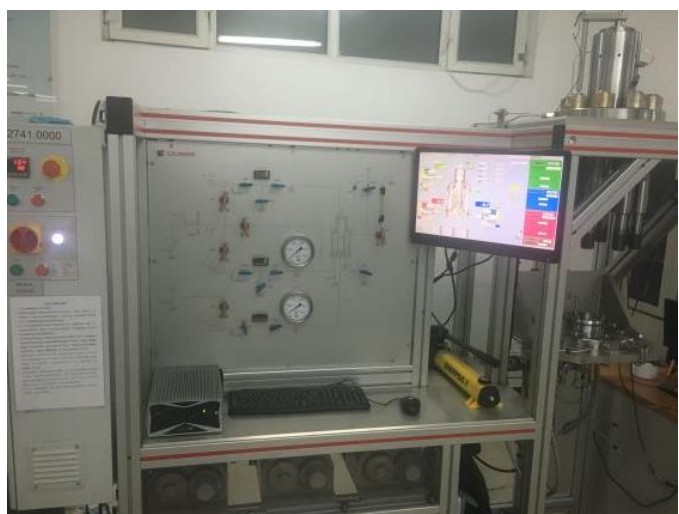

**Figure 1.** TOP INDUSTRIE Rock 600-50 Rock automatic servo rheometer.

### 2.2. Test Scheme

The unloading test adopted a constant-axial-pressure unloading confining pressure test; that is, under the premise of bearing both axial and confining pressure, the confining pressure is removed and the axial pressure remains unchanged until the specimen is destroyed. The specific steps are as follows: (1) according to the conditions of hydrostatic pressure, the confining pressure is applied to a predetermined value; (2) the confining pressure remains unchanged, and axial pressure is applied to 80% of the peak strength of the specimen; (3) keep the axial pressure is kept constant, and the confining pressure is removed until the specimen reaches total failure. The loading and unloading processes were controlled by stress. The confining pressure loading rate was $0.05 \text{ MPa·s}^{-1}$, and the axial pressure loading and confining pressure unloading rates were $0.05 \text{ MPa·s}^{-1}$. The predetermined confining pressure was set as 5, 10, 15, and 20 MPa, and each group was repeated three times. See Table 1 for the specific test scheme.

**Table 1.** Test scheme.

| Sample ID | Confining Pressure (MPa) | Peak Strength (MPa) |
|---|---|---|
| X1-1, X1-2, X1-3 | 5 | 96.86 |
| X2-1, X2-2, X2-3 | 10 | 120.02 |
| X3-1, X3-2, X3-3 | 15 | 146.04 |
| X4-1, X4-2, X4-3 | 20 | 161.21 |

## 3. Analysis of Crack-Characteristic Stress

### 3.1. Calculation Method of Crack Strain

In the process of rock deformation and failure, microcracks in rock undergo the process of closure, initiation, propagation, and coalescence. The evolution of microcracks can be used as the mechanism of rock micro-damage, and the crack strain can be used as an index to describe the evolution of microcracks in rock. Crack strain refers to the axial and lateral deformation caused by crack change under external load; those changes contain the initiation, propagation, and coalescing of primary cracks and the initiation of new cracks under external loads [16]. In the test process, the volumetric strain of rock consists of the

volumetric strain and elastic strain of the crack formed by the closure of the primary crack or the initiation and propagation of the new crack, which can be expressed as follows [17–19]:

$$\varepsilon_v = \varepsilon_v^e + \varepsilon_v^c \tag{1}$$

where $\varepsilon_v^e$ and $\varepsilon_v^c$ are the elastic volumetric strain and crack volumetric strain, respectively.

According to Hooke's law, the elastic volume strain is

$$\varepsilon_v^e = \varepsilon_1^e + \varepsilon_2^e + \varepsilon_3^e = \frac{1-2v}{E}(\sigma_1 + \sigma_2 + \sigma_3) \tag{2}$$

where $\varepsilon_1^e$, $\varepsilon_2^e$, and $\varepsilon_3^e$ are the elastic strain energies corresponding to the principal stress, respectively, and $\sigma_1$, $\sigma_2$ and $\sigma_3$ are the maximum principal stress, intermediate principal stress, and minimum principal stress, respectively; $E$ and $v$ are the elastic modulus and Poisson's ratio, respectively.

During the test, the volumetric strain can be obtained by $\varepsilon_1$, $\varepsilon_2$, and $\varepsilon_3$; its expression is as follows [17–19]:

$$\varepsilon_v = \varepsilon_1 + \varepsilon_2 + \varepsilon_3 \tag{3}$$

where $\varepsilon_1$, $\varepsilon_2$, and $\varepsilon_3$ are the strains corresponding to the directions of maximum principal stress, intermediate principal stress, and minimum principal stress, respectively.

According to Equations (1)–(3), the volume strain of the crack is

$$\varepsilon_v^c = \varepsilon_v - \varepsilon_v^e = \varepsilon_1 + \varepsilon_2 + \varepsilon_3 - \frac{1-2v}{E}(\sigma_1 + \sigma_2 + \sigma_3) \tag{4}$$

In the conventional triaxial loading and unloading test, the confining pressure is applied in the circular direction; thus, the above equation can be simplified as

$$\varepsilon_v^c = \varepsilon_1 + 2\varepsilon_3 - \frac{1-2v}{E}(\sigma_1 + 2\sigma_3) \tag{5}$$

*3.2. Analysis of Stress–Strain Curve*

According to the above calculation method, each strain of the rock in the process of unloading confining pressure is calculated, and the stress–strain curve is drawn. Due to space limitations, taking the confining pressure of 10 MPa as an example, the stress-strain curves and variation trends of each strain of rock samples were analyzed. Figure 2 represents the stress–strain curves of sandstone samples. The stress–strain change during confining pressure unloading is divided into five stages:

(1) Compaction stage (o–$\sigma_{cc}$ stage): the axial strain increases slowly and presents a concave growth. The growth of circumferential strain is small, almost zero. The volume strain trend and value are consistent with the axial strain. The volume strain of the crack gradually decreases to zero, and the primitive micro-cracks gradually close at this stage.

(2) Linear elastic deformation stage ($\sigma_{cc}$–$\sigma_{ci}$ stage): the starting and ending positions correspond to the crack closure stress $\sigma_{cc}$ and crack initiation stress $\sigma_{ci}$, respectively. The axial strain increases linearly, the circumferential strain increases slightly, and the volumetric strain increases approximately linearly. The volumetric strain of the crack is zero. At this stage, the rock sample gradually compacts and begins to crack.

(3) Steady crack growth stage ($\sigma_{ci}$–$\sigma_{cd}$ stage): when the crack initiation stress $\sigma_{ci}$ is exceeded, the axial strain still keeps a large growth rate, the growth rate of circumferential strain increases significantly, the volume strain increases slightly, reaching the maximum, and the volume strain of crack begins to increase. At this stage, rock sample cracks develop steadily and increase continuously.

(4) Expansion cracking development stage ($\sigma_{cd}$–$\sigma_p$ stage): when the dilatancy stress $\sigma_{cd}$ is exceeded, the axial strain and circumferential strain increase stably, the volume strain deflects to the negative direction, the volume strain decreases rapidly, and the crack

volume strain increases obviously. At this stage, the volumetric strain is reversed, the volume compression decreases obviously, and the rock samples enter the plastic yield stage.

(5) Post peak stage (the stage after $\sigma_p$): when the peak stress $\sigma_p$ is exceeded, almost axial strain, circumferential strain, and volume strain increase rapidly, crack volume strain also increases significantly, and the sample volume expands. The reason is the rapid development of internal cracks in rock samples, forming a large area of penetration, resulting in significant volume expansion; rock samples also undergo failure.

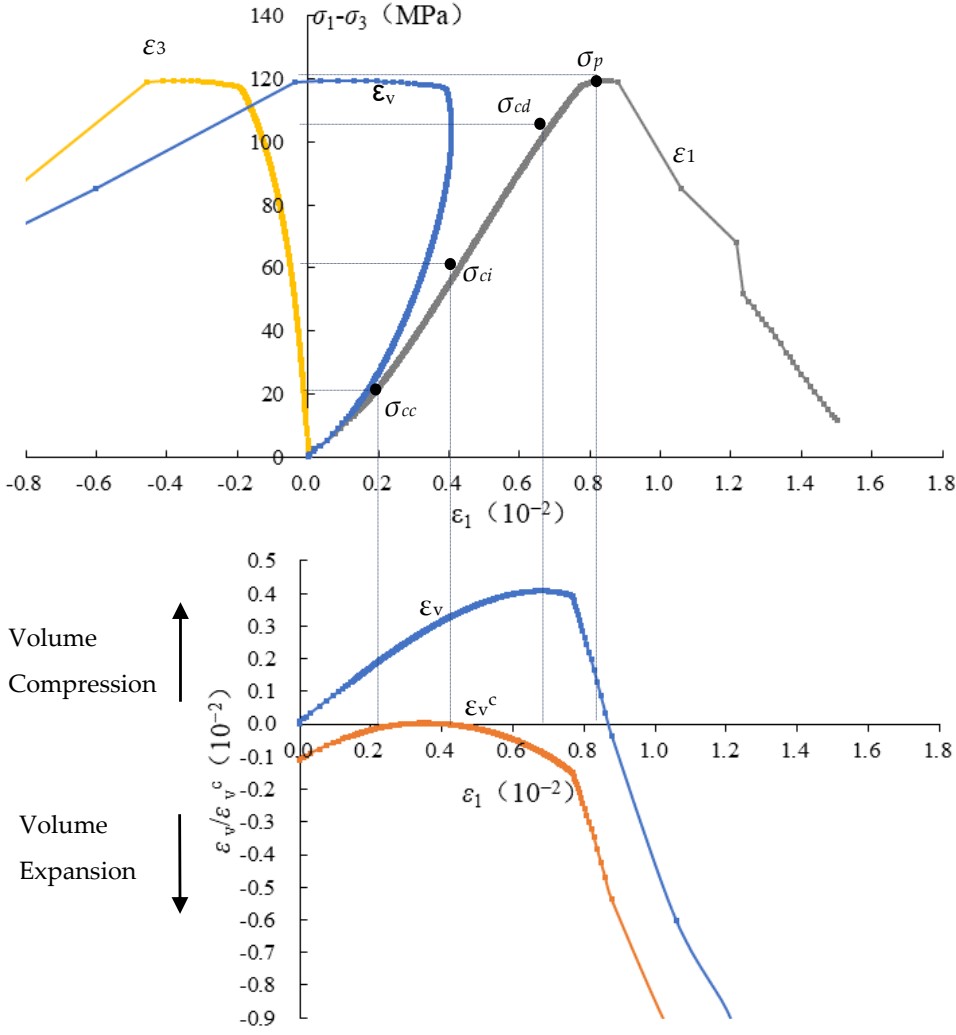

**Figure 2.** Stress–strain curves of sandstone ($\sigma_3 = 10$ MPa).

### 3.3. Analysis of Crack-Characteristic Stress

According to the content in the above section, each crack-characteristic stress is shown in Table 2. When the confining pressure increases from 5 MPa to 20 MPa, the crack closure stress $\sigma_{cc}$ increases from 12.55 MPa to 79.52 MPa, which is 6.34 times the original; the crack initiation stress $\sigma_{ci}$ increases from 44.71 MPa to 122.89 MPa, which is 2.75 times the original; the dilatancy stress $\sigma_{cd}$ increases from 79.61 MPa to 153.25 MPa, which is 1.93 times the original; the peak stress $\sigma_p$ increases from 96.86 MPa to 161.21 MPa, which is 1.66 times the original. By comparing the amplitude of crack-characteristic stress, it can be found that confining pressure greatly influences crack closure stress and crack initiation stress, but has relatively little influence on dilatation stress and peak stress. The reason is that the confining pressure limits the closure of the primary crack and the initiation of new cracks

in the rock sample to a certain extent. However, when the dilatancy stress is reached, the rock sample enters the plastic yield state, and its deformation and damage are great.

**Table 2.** Crack-characteristic stress.

| $\sigma_3$ (MPa) | $\sigma_{cc}$ (MPa) | $\sigma_{ci}$ (MPa) | $\sigma_{cd}$ (MPa) | $\sigma_p$ (MPa) | $\sigma_{cc}$ ($\sigma_p$) | $\sigma_{ci}$ ($\sigma_p$) | $\sigma_{cd}$ ($\sigma_p$) |
|---|---|---|---|---|---|---|---|
| 5 | 12.55 | 44.71 | 79.61 | 96.86 | 0.130 | 0.462 | 0.822 |
| 10 | 21.62 | 61.82 | 106.06 | 120.02 | 0.180 | 0.515 | 0.884 |
| 15 | 55.85 | 103.56 | 130.91 | 146.04 | 0.382 | 0.709 | 0.896 |
| 20 | 79.52 | 122.89 | 153.25 | 161.21 | 0.493 | 0.762 | 0.951 |

Figures 3 and 4 show the crack-characteristic stress and ratio of crack-characteristic stress of sandstone under different confining pressures, respectively. The crack-characteristic stress increases linearly, reflecting the obvious confining pressure effect. The crack-characteristic stress ratio increases with different amplitude; the ratio of $\sigma_{cc}$ to $\sigma_p$ greatly increases from 0.130 to 0.493, and the ratio $\sigma_{ci}$ to $\sigma_p$ greatly increases from 0.462 to 0.762. The ratio of $\sigma_{cd}$ to $\sigma_p$ increases from 0.822 to 0.951, which is a large value with a small increase. The variation law of the crack-characteristic stress ratio is consistent with the crack-characteristic stress. This indicates that the crack-characteristic stress has a good linear relationship with the confining pressure.

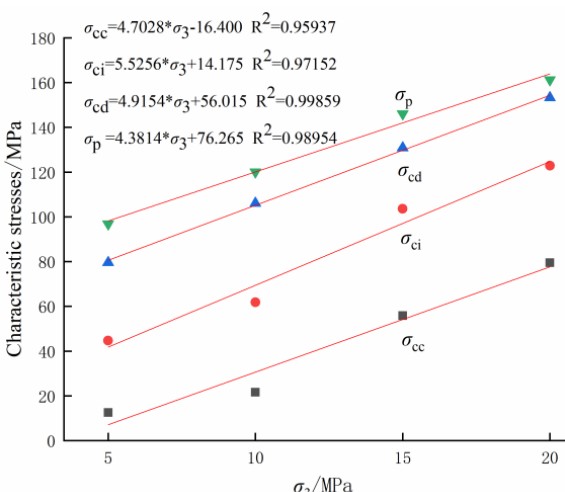

**Figure 3.** Crack-characteristic stresses of sandstone under different confining pressure.

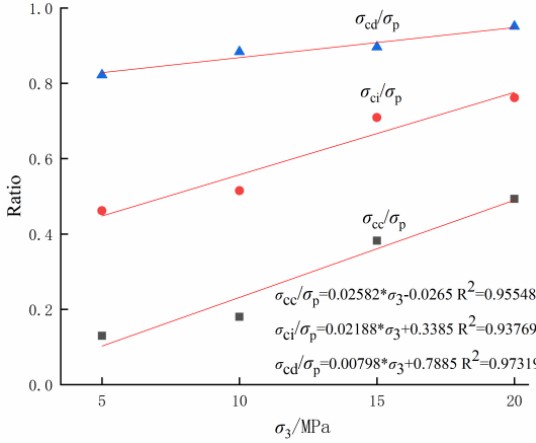

**Figure 4.** The ratio of crack-characteristic stresses of sandstone under different confining pressure.

## 4. Analysis of Energy Evolution Characteristics of Triaxial Unloading Confining Sandstone

### 4.1. Energy Calculation Method

Under the action of the mechanical testing machine, the energy inside the rock is in different states; in the initial loading stage, the mechanical energy and heat energy are input into the rock, the rock undergoes elastic deformation, and the energy can be stored in the rock. The whole process is dynamically balanced. As the loading goes on, the deformation capacity of rock becomes worse, and the elastic energy that can be stored becomes less. During this period, external input energy causes new micro-cracks in the rock. Although the energy at this stage is in dynamic equilibrium, it is extremely unstable. When the external load is too large, the input energy causes the internal microscopic cracks of the rock to be connected, and the macroscopic cracks begin to appear. When the energy accumulated in the rock exceeds its energy storage limit, the energy is released instantaneously, resulting in rock damage, and the dynamic image of rock block flying may be caused. When the energy is released, the rock returns to a new equilibrium state [20–22].

In the process of rock specimens being loaded, energy storage and dissipation are always present. The whole energy evolution process can be expressed in terms of elastic energy and dissipative energy. In laboratory experiments, these variables are not easily obtained directly and can be calculated from real-time recorded stress–strain curves [23–26]. Figure 5 shows the calculation principle of energy. Taking the peak stress point as an example, Eu is the unloading modulus, elastic energy $U^e$ is the shadow area, and dissipated energy $U^d$ is the area enclosed by the stress–strain curve, unloading modulus, and transverse axis.

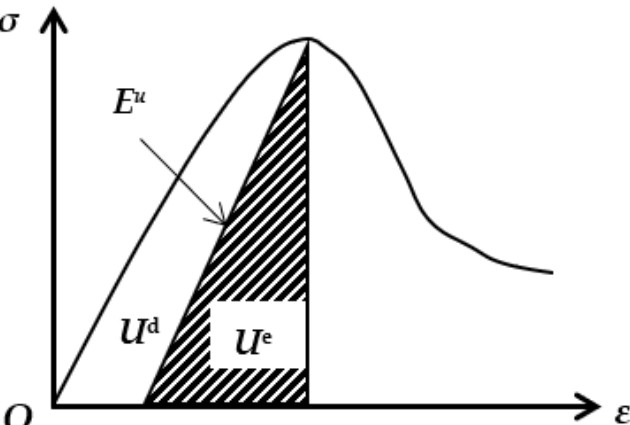

**Figure 5.** Calculation principle of energy.

According to the first law of thermodynamics, the energy $U$ generated by the external forces is as follows [20,23–26]:

$$U = U^e + U^d, \tag{6}$$

where $U^e$ is elastic energy, and $U^d$ is dissipated energy.

It is assumed that the system is a closed and the heat transfer during the test is not taken into account, whereby all the mechanical energy generated by the testing machine enters the rock sample, i.e., $U = U_0$. In the triaxial compression test, the axial absorption energy $U_1$ is positive, and the circumferential consumption energy $U_3$ is negative. The rock absorption energy $U_0$ is equal to the sum of axial absorption energy $U_1$ and circumferential consumption energy $U_3$, i.e.,

$$U_0 = U_1 + U_3. \tag{7}$$

The axial absorbed energy $U_1$ and circumferential consumption energy $U_3$ of rock can be represented as follows [18,20]:

$$U_1 = \int \sigma_1 d\varepsilon_1 = \sum_{i=0}^{n} \frac{1}{2}(\varepsilon_{1i+1} - \varepsilon_{1i})(\sigma_{1i} + \sigma_{1i+1}) \tag{8}$$

$$U_3 = 2\int \sigma_3 d\frac{\varepsilon_3}{2} = \sum_{i=0}^{n} \frac{1}{2}(\varepsilon_{3i+1} - \varepsilon_{3i})(\sigma_{3i} + \sigma_{3i+1}) \tag{9}$$

where $\sigma_{1i}$, $\varepsilon_{1i}$, $\sigma_{3i}$, and $\varepsilon_{3i}$ are the axial stress, axial strain, circumferential stress, and circumferential strain, respectively.

The elastic energy of triaxial compression can be expressed as follows [18,19]:

$$U^e = \frac{1}{2E^u}[\sigma_1^2 + 2\sigma_3^2 - 2\mu(2\sigma_1\sigma_3 + \sigma_2\sigma_3)] \tag{10}$$

where $E^u$ is the elastic modulus when unloading, and $\mu$ is 50–60% of the peak strength [19,20,25].

### 4.2. Energy Evolution Analysis

According to the above calculation principles, the relationship between the energy evolution characteristics and the stress–strain curve was drawn, as shown in Figure 6.

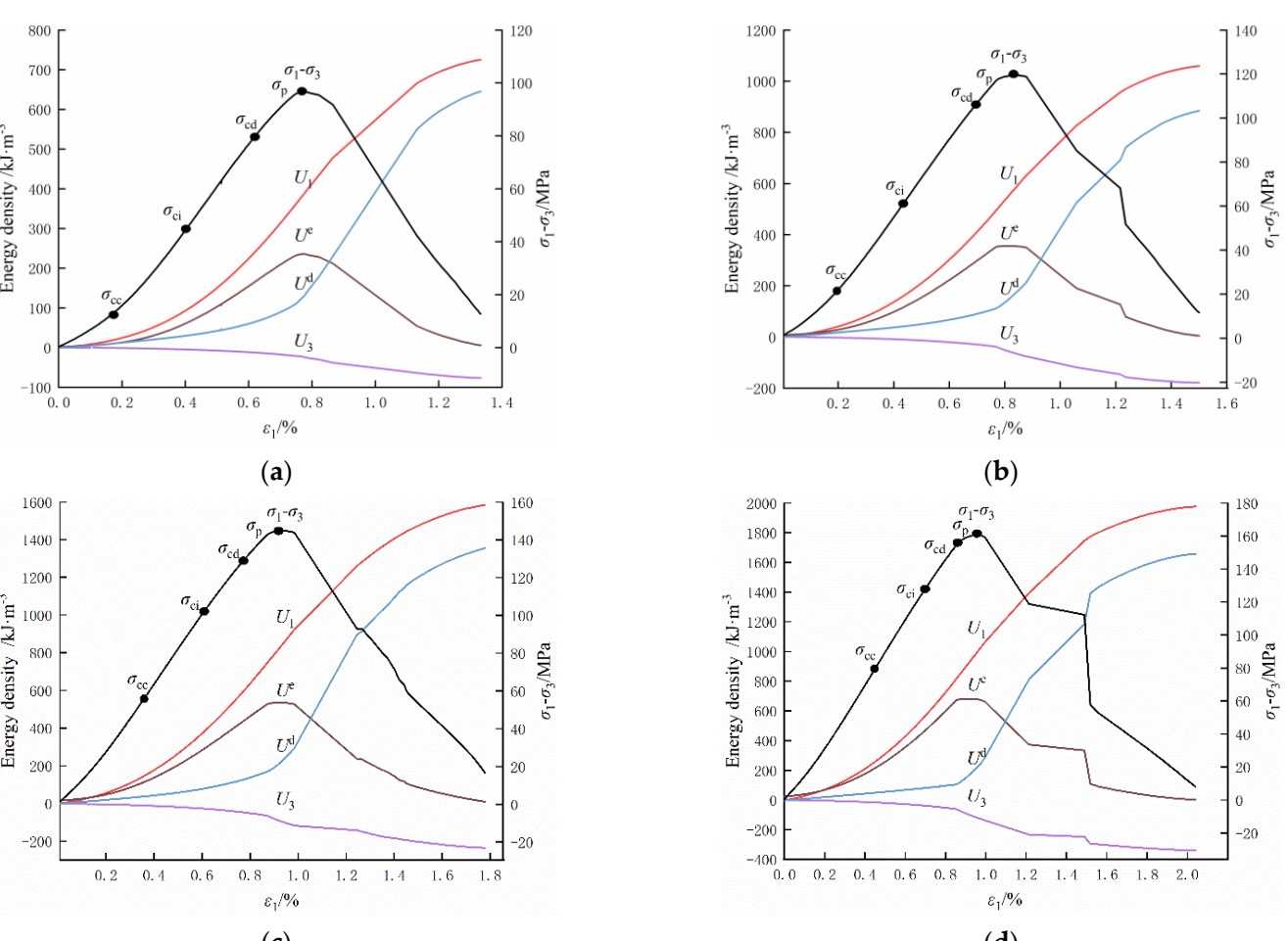

**Figure 6.** Energy evolution curve of sandstone under unloading confining pressure: (**a**) $\sigma_3 = 5$ MPa, (**b**) $\sigma_3 = 10$ MPa, (**c**) $\sigma_3 = 15$ MPa, and (**d**) $\sigma_3 = 20$ MPa.

As can be seen from Figure 6, the evolution of the axial absorption energy $U_1$ is similar under different confining pressures. With the increase in axial strain, $U_1$ increases slowly before the crack closure stress $\sigma_{cc}$. Before reaching the peak stress $\sigma_p$, the growth rate increases. However, after $\sigma_p$, the growth rate gradually slows down. The overall growth rate is a slow–fast–slow process, which is related to the characteristics of the stress–strain curve. The maximum axial absorbed energy $U_1$ increases with the increase in confining pressure.

Different confining pressures have similar evolutions to $U_3$ energy consumption. With the increase in axial strain, $U_3$ shows a different growth trend before and after peak stress. The growth is slow before $\sigma_p$, especially in the crack closure stress $\sigma_{cc}$, while the growth is rapid after the peak stress. $U_3$ is positively correlated to the confining pressure.

The evolution law of elastic energy $U^e$ is consistent with the stress–strain curve. Before the peak stress, $U^e$ accounts for a large proportion of $U_1$, while $U_3$ and $U^d$ are small. After reaching $\sigma_p$, $U^e$ decreases rapidly, while $U_3$ and $U^d$ increase rapidly, and $U^d$ rapidly becomes the main energy. As the radial strain increases, the difference between $U^e$ and $U_1$ becomes larger and larger, and the $U^e$ in peak stress is about one-third of that of $U_1$. $U^e$ is positively correlated with confining pressure.

The evolution trend of the dissipated energy $U^d$ is similar to that of $U_3$, with slow growth before $\sigma_p$ but rapid growth after $\sigma_p$. The difference is that $U_3$ is a very small part of the total energy; thus, it is always small, while $U^d$ grows almost exponentially after $\sigma_p$ and rapidly becomes the main part of the energy. The maximum of $U^d$ is different and increases with confining pressures. After $\sigma_p$, the $U^d$ growth rate increases slightly.

### 4.3. Analysis of Pre-Peak Energy Characteristics

According to the energy evolution curve of sandstone unloading confining pressure under different confining pressures in Figure 6, the energy eigenvalue under the state of peak stress is statistically shown in Table 3. In addition, the relationship between confining pressure and energy eigenvalue is analyzed according to the energy eigenvalue in Table 2. Figure 7 shows the variation curves of energy eigenvalue under different confining pressures, and the absolute value of circumferential energy consumption $U_3$ is taken.

**Table 3.** Energy characteristic parameters under different characteristic stresses.

| Crack-Characteristic Stress | $\sigma_3$ (MPa) | Energy Characteristic Parameters | | | |
|---|---|---|---|---|---|
| | | $U_1$ (kJ·m$^{-3}$) | $U_3$ (kJ·m$^{-3}$) | $U^e$ (kJ·m$^{-3}$) | $U^d$ (kJ·m$^{-3}$) |
| $\sigma_{cc}$ | 5 | 18.04 | −0.66 | 9.71 | 9.91 |
| | 10 | 39.65 | −2.50 | 27.52 | 17.00 |
| | 15 | 146.13 | −10.33 | 113.89 | 37.53 |
| | 20 | 256.85 | −16.63 | 213.73 | 48.63 |
| $\sigma_{ci}$ | 5 | 94.42 | −4.73 | 61.75 | 30.18 |
| | 10 | 163.32 | −10.29 | 118.53 | 41.87 |
| | 15 | 391.10 | −27.92 | 297.26 | 74.19 |
| | 20 | 526.80 | −35.68 | 436.40 | 84.21 |
| $\sigma_{cd}$ | 5 | 239.16 | −12.44 | 164.59 | 64.36 |
| | 10 | 405.93 | −28.90 | 295.16 | 89.25 |
| | 15 | 614.90 | −48.75 | 449.66 | 114.52 |
| | 20 | 801.76 | −59.78 | 660.09 | 117.48 |
| $\sigma_p$ | 5 | 390.47 | −24.25 | 235.59 | 132.87 |
| | 10 | 580.21 | −63.26 | 355.21 | 169.12 |
| | 15 | 848.59 | −97.90 | 536.27 | 206.96 |
| | 20 | 1003.50 | −122.07 | 678.06 | 225.52 |

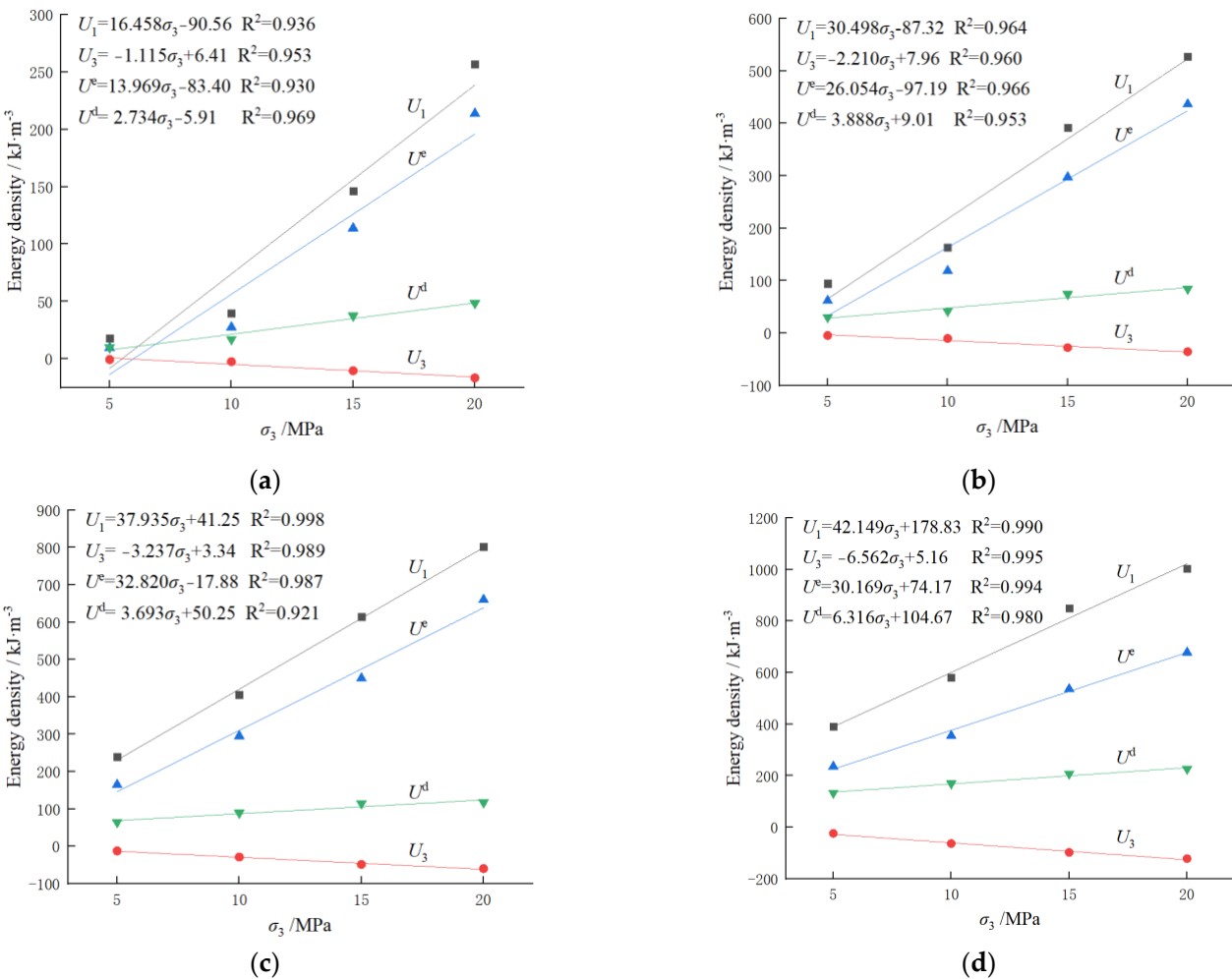

**Figure 7.** Energy eigenvalue curves under different confining pressure: (**a**) $\sigma_{cc}$, (**b**) $\sigma_{ci}$, (**c**) $\sigma_{cd}$, and (**d**) $\sigma_p$.

As can be seen from Table 3 and Figure 7, under each crack-characteristic stress state, each energy characteristic value has a linear relationship with the confining pressure, reflecting an obvious confining pressure effect. Under the crack closure stress $\sigma_{cc}$ state, the axial absorption energy $U_1$ increases the most, from 18.04 kJ·m$^{-3}$ to 256.85 kJ·m$^{-3}$, with an increase of 238.18 kJ·m$^{-3}$. This is followed by the increase in elastic energy $U^e$, which increases from 9.71 kJ·m$^{-3}$ to 213.73 kJ·m$^{-3}$, with an increase of 240.02 kJ·m$^{-3}$. The value of circumferential energy consumption $U_3$ is negative, with the smallest value increasing from 0.66 kJ·m$^{-3}$ to 16.63 kJ·m$^{-3}$. The dissipation energy $U^d$ increases slightly more than that of $U_3$, from 9.91 kJ·m$^{-3}$ to 48.63 kJ·m$^{-3}$, with an increase of 38.72 kJ·m$^{-3}$. The change value of energy consumption relative to elastic energy is very small, which indicates that the energy used for the internal consumption of rock is very small. As the rock sample is destroyed, most of the stored energy is released. In the subsequent states of $\sigma_{ci}$, $\sigma_{cd}$, and $\sigma_p$, the increased amplitude gradually increases, which is shown by the increase in slope in the linear fitting formula of energy characteristic parameters. For example, the slope of the $U_1$ fitting formula increases from 16.458 under the crack closure stress state to 42.149 under the peak stress state, which shows that the values of each energy characteristic parameter gradually increase before the peak stress, reflecting a strong confining pressure effect.

### 4.4. Analysis of Post-Peak Energy Release Characteristics

Energy release refers to the elastic energy that can be releasable after peak stress, which is directly related to unloading elastic modulus and unloading Poisson's ratio. From the

perspective of thermodynamics, energy dissipation is unidirectional and irreversible, while energy release is bidirectional and reversible as long as certain conditions are met [26–28]. Therefore, the change in elastic energy after peak stress was studied.

Figure 8 shows the variation curves of releasable elastic energy under different confining pressures. The release process of post-peak releasable elastic energy can be divided into stage I, stage II and stage III. In stage I, the release rate is the slowest and the axial strain variation range is the smallest, which is 0.05%. Stage II has the fastest release rate and the largest axial strain variation range. With confining pressure increasing from 5 MPa to 20 MPa, energy reduction increases from 172.92 kJ·m$^{-3}$ to 555.01 kJ·m$^{-3}$, and axial strain increases from 1.132% to 1.52%. In stage III, the release rate is slow and the variation range of axial strain is small, which is basically 0.2–0.5%. In conclusion, the energy release process shows an obvious confining pressure effect.

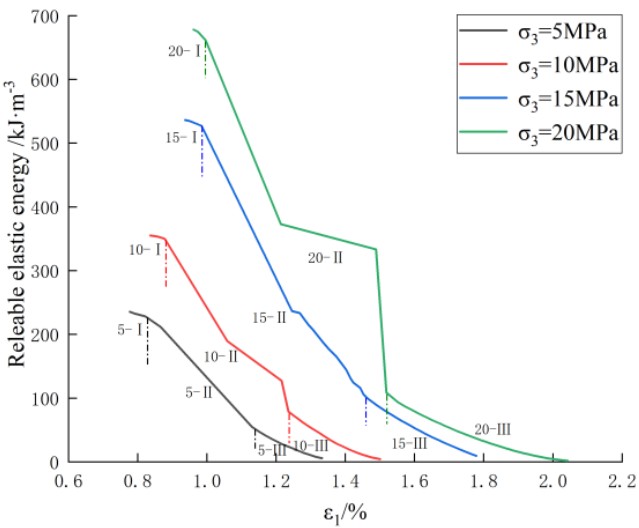

**Figure 8.** Variation curves of reliable elastic energy under different confining pressures.

## 5. Discussion

(1) There is no description of crack morphology for the analysis of crack characteristic stress. For example, what is the crack morphology under the state of crack initiation stress? How is it distributed? Most of the research [29–32] obtained mechanical properties through indoor tests, and then used numerical simulation software to study the distribution law of crack morphology. In addition, the tests carried out in this study were limited; hence, the number and size of the test groups should be increased, and other unloading schemes should be investigated to supplement the unloading conditions. Therefore, experimental research on the distribution law of crack morphology will be one of the key points of follow-up research.

(2) There is a lack of in-depth research on the characteristics of post-peak energy release. For example, the research process of released elastic energy after the peak stress can be discussed at the post-peak stage to study the characteristics of each energy release stage. In addition, the energy release is only analyzed from the view of axial strain, lacking a comprehensive analysis. It should also be analyzed over time to determine the time characteristics of energy release. These two aspects will be the focus of future research.

## 6. Conclusions

In this work, the triaxial pressure unloading confining pressure test was carried out to simulate the failure process of rock mass caused by unloading, analyze the mechanical characteristics of the deformation and failure process, and study the energy evolution and

energy release law of rock under the unloading action combined with the energy theory. The conclusions are as follows:

1.  When the confining pressure increased from 5 MPa to 20 MPa, the crack closure stress $\sigma_{cc}$, crack initiation stress $\sigma_{ci}$, dilatancy stress $\sigma_{cd}$, and peak stress $\sigma_p$ were 6.34 times, 2.75 times, 1.93 times, and 1.66 times higher than the original, respectively. By comparing the increase in characteristic stress, it was found that the confining pressure had a great influence on the crack closure stress and crack initiation stress, but relatively little influence on the dilatation stress and peak stress.

2.  Before the peak stress, the axial absorbed energy $U_1$ increased exponentially, the elastic energy $U^e$ was similar to $U_1$, and the circumferential consumption energy $U_3$ and dissipated energy $U^d$ were small. After reaching the peak stress, the growth rate of $U_1$ decreased slightly, $U^e$ decreased rapidly, and $U_3$ increased rapidly but only accounted for a small proportion of the total energy, while $U^d$ grew almost exponentially and rapidly became the main part of the energy. Each energy eigenvalue increased linearly, reflecting an obvious confining pressure effect; axial absorption energy $U_1$ increased the most, followed by elastic energy $U^e$, while annular consumption energy $U_3$ increased the least, and dissipation energy $U^d$ increased slightly more than $U_3$.

3.  Under each crack-characteristic stress state, each energy characteristic parameter increased with the increase in confining pressure, as shown by the increase in slope in the linear fitting formula of the energy characteristic parameter. For example, the slope of the $U_1$ fitting formula increased from 16.458 under the crack closure stress state to 42.149 under the peak stress state, indicating that the value of each energy characteristic parameter increased gradually before the peak stress, reflecting a strong confining pressure effect.

4.  The post-peak elastic energy release process could be divided into three stages: "slow–fast–slow". In the first stage, the release rate was the slowest and the axial strain variation range was the smallest, at basically 0.05%. The second stage had the fastest release rate and the largest axial strain variation range. In the third stage, the release rate was slow, and the variation range of axial strain was small. The energy release process showed an obvious confining pressure effect.

**Author Contributions:** Conceptualization, G.Z.; validation, Y.D.; formal analysis and investigation, T.Q. and Y.D.; data curation, Y.D. and T.Q.; writing—original draft preparation, T.Q. and Y.D.; writing—review and editing, G.Z. and Y.D.; supervision, G.Z. All authors have read and agreed to the published version of the manuscript.

**Funding:** This work was supported by the Scientific and Technological Key Project of "Revealing the List and Taking Command" in Heilongjiang Province: Study on geological model and ventilation model of intelligent mining in extremely thin coal seam (2021ZXJ02A03).

**Institutional Review Board Statement:** Not applicable.

**Informed Consent Statement:** Not applicable.

**Data Availability Statement:** Not applicable.

**Conflicts of Interest:** The authors declare no conflict of interest.

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
