# Peer review of "Analysis of Crack-Characteristic Stress and Energy Characteristics of Sandstone under Triaxial Unloading Confining Pressure"

_applsci, doi:10.3390/app13042671_

Round 1

Reviewer 1 Report

The article raises an important issue of assessment and energy characteristics of Carboniferous sandstones, which is a useful cognitive element in the assessment of rock mass deformation processes caused by mining.

However, the article is not free from some errors that I have marked in the text of the article. In addition, the reviewer has several doubts:

1. The reviewer is unsatisfied with the number of samples subjected to laboratory tests. From the text it can be deduced that the research was carried out on 4 sandstone samples (collected probably from one site), and each of these samples was divided into 3 smaller ones. It seems that this is a bit too small a number of samples to draw meaningful conclusions from them. Here, I would like to ask for the authors' comments in the appropriate place in the text.

2. I have doubts about the linear interpretation for all energy functions. I have included the comments in the text.

3. I found a lot of editorial errors in the text, especially in the correct notation of symbols - I have included comments in the text.

Author Response

Point 1: The reviewer is unsatisfied with the number of samples subjected to laboratory tests. From the text it can be deduced that the research was carried out on 4 sandstone samples (collected probably from one site), and each of these samples was divided into 3 smaller ones. It seems that this is a bit too small a number of samples to draw meaningful conclusions from them. Here, I would like to ask for the authors' comments in the appropriate place in the text.

Response 1: The specimens used in this manuscript are all from the roof of the left fourth working face of the Xinjian Coal mine in Qitaihe, with the same lithology. The confining pressure of the test was set to 4 groups and each group was repeated three times. It provides the experimental basis for studying crack-characteristic stress and energy characteristics of rock under this lithology. Of course, due to the limitation of research conditions, this test scheme has some shortcomings. A wider range of test contents can be studied, such as lithologies, rocks with weak surfaces, rock sample sizes, and stress paths.

Point 2: I have doubts about the linear interpretation for all energy functions. I have included the comments in the text.

Response 2: Re-proofread the test data and make the corresponding modification. According to the current data fitting effect, the energy function presents a linear relationship. Of course, because only 4 groups of confining pressure are set, which is relatively few, other functional forms are also fitted. The R2 values of the fitting indexes in the manuscript are all above 0.9, so it can be considered that the energy function has a linear relationship with the confining pressure.

Point 3: I found a lot of editorial errors in the text, especially in the correct notation of symbols - I have included comments in the text.

Response 3: The text's editing errors and symbol format problems have been corrected, and the modified parts are highlighted in yellow. Other issues noted in the manuscript have also been revised accordingly.

Reviewer 2 Report

The manuscript presents a study on sandstone's crack characteristic stress and energy components under triaxial unloading confining pressure. A series of triaxial confining pressure tests was conducted to investigate the deformation and failure process of rock mass and study the energy evolution and energy release law of rock under the unloading combined with the energy theory. The expected outcomes of the study would potentially be helpful for readership.

I find the study interesting, and the results will be helpful for the readership. I provide my comments below for the authors’ consideration for further improving the quality of the revised manuscript.

·         Introduction: I suggest the authors carry out a more critical review and discussion on related studies, particularly on energy components of rock mass subjected to loading-unloading process.

·         Page 2, Paragraph 3: The authors should also emphasise a research gap, a need for motivation and the original contribution of the current study.

·         Figures 1 & 5: References to their sources are recommended. Typo error in Figure 5 caption.

·         Equations 1-5 are well established, and I suggest to remove them or include an original reference.

·         Figures 3 & 4: it is unclear to me how the cracks were monitored and recorded during the test. Please elaborate on this matter.

·         Equations 9, 10, 11: I am not convinced how these Equations were derived. Please provide more detailed information or sources.

·         Figure 8: there is a lack of data interpretation for this Figure.

·         Figure 6: provide detailed information on how the energy density is calculated.

·         Some references are in the Chinese language. I suggest to include alternative references in English for the benefit of readership.

·         Equations 7, 8 and 10: a reference to its source is recommended.

·         I suggest including a section on the practical implications of the current study, focusing on how practising engineers can potentially adopt the results of this study.

Author Response

Point 1: Introduction: I suggest the authors carry out a more critical review and discussion on related studies, particularly on energy components of rock mass subjected to loading-unloading process.

Response 1: In the Introduction part, the research status of energy characteristics of rock mass in the loading process is reviewed.

Point 2: Page 2, Paragraph 3: The authors should also emphasise a research gap, a need for motivation and the original contribution of the current study.

Response 2: In the Introduction part, it added the research gap, motivation need and original contribution of current research.

Point 3: Figures 1 & 5: References to their sources are recommended. Typo error in Figure 5 caption.

Response 3: Added relevant references for Figures 1 and 5, and corrected the caption error in Figure 5.

Point 4: Equations 1-5 are well established, and I suggest to remove them or include an original reference.

Response 4: The necessary reference sources for Equations 1-5 were added to the manuscript.

Point 5: Figures 3 & 4: it is unclear to me how the cracks were monitored and recorded during the test. Please elaborate on this matter.

Response 5: The crack characteristic stresses in Figure 3 and Figure 4 are reflected by the stress-strain curve, and the crack characteristic stress point corresponds to the stress mutation point during the test.

Point 6: Equations 9, 10, 11: I am not convinced how these Equations were derived. Please provide more detailed information or sources.

Response 6: The necessary reference sources for Equations 9-11 were added to the manuscript.

Point 7: Figure 8: there is a lack of data interpretation for this Figure

Response 7: The relevant data interpretation in Figure 8 was added to the manuscript.

Point 8: Figure 6: provide detailed information on how the energy density is calculated.

Response 8: The energy density in Figure 6 is calculated by Equations 6-10. Due to the large unit interval in Figure 6, the energy density value is not reflected in the figure, which can be seen in detail in Table 3.

Point 9: Some references are in the Chinese language. I suggest to include alternative references in English for the benefit of readership.

Response 9: English references were added to the manuscript where appropriate.

Point 10: Equations 7, 8 and 10: a reference to its source is recommended.

Response 10: The necessary reference sources for Equations 7,8 and 10 were added to the manuscript.

Point 11: I suggest including a section on the practical implications of the current study, focusing on how practising engineers can potentially adopt the results of this study.

Response 11: In the Introduction part, the application prospect and research significance of the research results are added.

Round 2

Reviewer 2 Report

The authors have adequately attended to the comments and suggestions provided by this reviewer. I have now endorsed the revised manuscript for publication.